# RetriBooru: Leakage-Free Retrieval of Conditions from Reference Images for Subject-Driven Generation

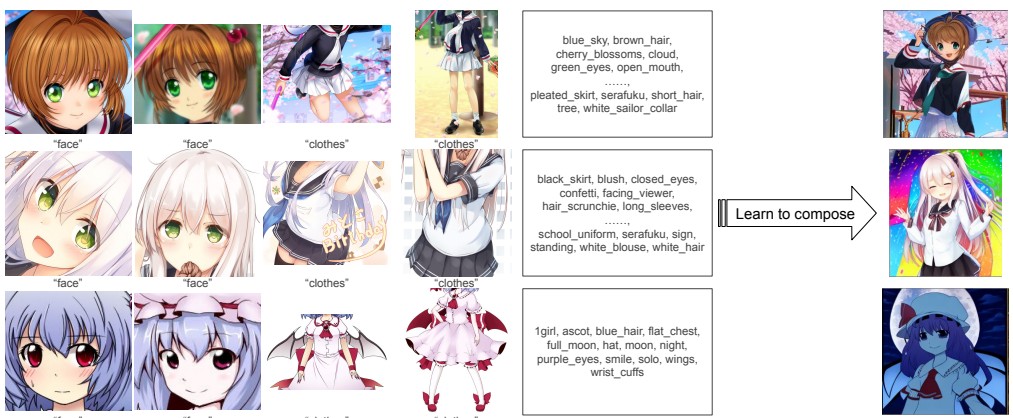

Figure 1: Training the proposed concept composition task on our RetriBooru dataset. Different concepts to retrieve are specified in texts and passed to the retrieval encoder, which learns only from characteristic information to compose the output, guiding generation with text prompts in the U-Net.

## Abstract

Diffusion-based methods have demonstrated remarkable capabilities in generating a diverse array of high-quality images, sparking interests for styled avatars, virtual try-on, and more. Previous methods use the same reference image as the target. An overlooked aspect is the leakage of the target's spatial information, style, etc. from the reference, harming the generated diversity and causing shortcuts. However, this approach continues as widely available datasets usually consist of single images not grouped by identities, and it is expensive to recollect large-scale same-identity data. Moreover, existing metrics adopt decoupled evaluation on text alignment and identity preservation, which fail at distinguishing between balanced outputs and those that over-fit to one aspect. In this paper, we propose a multi-level, same-identity dataset RetriBooru, which groups anime characters by both face and cloth identities. RetriBooru enables adopting reference images of the same character and outfits as the target, while keeping flexible gestures and actions. We benchmark previous methods on our dataset, and demonstrate the effectiveness of training with a reference image different from target (but same identity). We introduce a new concept composition task, where the conditioning encoder learns to retrieve different concepts from several reference images, and modify a baseline network RetriNet for the new task. Finally, we introduce a novel class of metrics named Similarity Weighted Diversity (SWD), to measure the overlooked diversity and better evaluate the alignment between similarity and diversity.

## 1 Introduction

Recent advancements in latent diffusion models (Rombach et al., 2022) have ushered in new capabilities in image generation, including face generation, virtual try-on, anime creation, etc. Previously, the focus was on optimization-based methods which involved tuning entire networks (Ruiz et al., 2023), tuning low-rank matrices (Hu et al., 2021), or incorporating text embeddings (Gal et al., 2022). Yet, they suffer from slow inference and their hyper-parameters do not generalize to all intended subjects. The latest research has shifted towards injecting specific subject information directly into

the generation process, enhancing the relevance and accuracy of the output. Current methods can be categorized into two main approaches. The first approach, exemplified by IP-Adapter (Ye et al., 2023), integrates additional networks that do not specifically learn to comprehend reference images. The second approach, e.g. InstantID (Wang et al., 2024b), adopts specialized inputs such as facial key-points, which while effective for maintaining facial identity, lacks versatility in other contexts like changing outfits. However, we observe a potential defect of the predominant training pipeline, which adopts the same image for both reference conditioning and target. This setup can inadvertently leak target information to the conditioning encoder during training, which could lead to shortcut learning, e.g., spatial information (Xue et al., 2023; Zhong et al., 2023). While this pipeline may aid in convergence and stabilize generation, it raises concerns about whether the model is learning the intended concepts or merely exploiting the leaked information. In this paper, we propose a shift from learning a passive condition encoder to a retrieval encoder, enabled by new annotations. This new task encourages the encoder to understand and actively retrieve subject concepts in the intended manner, without depending on non-characteristic target information.

**Avoiding target leakage.** To learn to retrieve the specified information and reduce leakage from the target, a good conditional image could be chosen to maximize the shared characteristic information we would like to learn, and minimize the unwanted information. For example, vision-language models are used to be challenged by distinguishing between Chihuahua and Muffins (Fan et al., 2024), partially due to their similar spatial structures (near-round shapes in brown with black dots), which are heavily exploited during pre-training. If the non-characteristic information was well permuted, e.g. hiding Chihuahua partially, closing Chihuahua eyes, cutting muffins, resulting in a diverse and less learnable spatial inductive bias, we could force the models to learn from characteristic information as intended, such as furs on Chihuahua faces and cake crumbs on muffins, as well as their dog paws and paper cups, respectively. Applying the same ideology to subject-driven generation, what if we can access to conditional images with different spatial information, activities etc. from the target, while maintaining the identity? In this sense, the conditional encoder is forced to learn from the definitive concepts and features of the identity. However, such identity-preserving annotations are scarce. For human appearance, current identity-focused datasets are either limited to faces, or feature inconsistent clothing options. Meanwhile, virtual try-on datasets provide detailed clothing labels but suffer from a lack of diversity in body poses and background settings (Sec. 2). It remains a significant challenge to keep track of whole-body human identities with unlimited motions and poses, accounting for both facial and clothing identities, where the later one can lead the classification at whole-body scale too.

**Costs and generalization.** Another practical consideration is the trade-off between computational costs and the generalization ability. For instance, InstantID (Wang et al., 2024b) requires 48 NVIDIA H800 GPUs, with a batch size of only two, to train adapter structures for face generation with key-point features on their large, private dataset. Extending this to more general semantic features could significantly increase computational demands. Researchers, when validating prototype methods and primary results, demand a smaller and easier dataset, in order to obtain valuable feedback on the way to convergence. However, an easier benchmark would still require a decent ability to generalize, so that the observations and conclusions drawn can also apply to larger scale datasets and other domains. Taking both into consideration, anime figures data offers a valuable testing ground due to its easier details and good mimicry of human figures. It has less compressed latent space after VAE, which is less prone to degeneration at smaller resolutions compared to human faces and body parts. This allows for quicker iterations and faster algorithm development in anime styles, while fitting into tighter speed and memory constraints. On the other hand, anime figures are animated humans, whose samples share a large similarity in body structures, facial expressions, activities, etc. with human samples, facilitating human-related research. What is more, anime itself also serves as the medium of plenty artistic creations in books, movies and games, which have proven popular and successful for both commercial and recreational applications, sparking generative methods to create anime subjects.

Therefore, we propose RetriBooru, a multitask, multiconcept anime dataset based on Danbooru, one of the largest anime website with higher-quality anime images and precise tags. More importantly, We propose a labeling pipeline by leveraging existing tags and VQA models to construct same-outfit clusters for anime characters (Sec. 3). This combination of data and annotation would be very costly to obtain otherwise. With annotated clothing identities, RetriBooru not only enables training with a different reference image of the same identity as target, but also empowers more advanced tasks such as composing faces and clothes so changing one's outfits. In order to retrieve and inject desired information into the generation, we modify ControlNet (Zhang et al., 2023) to build a retrieval

encoder, namely RetriNet (Fig. 4). Simultaneous work follow a similar design, achieving good results in other generation tasks with other geometric or layout control (Sec. 2). We take this one step further and let both appearance and geometric information be retrieved and injected.

**Evaluating prompt-reference fusion.** In evaluating the generation quality, common CLIP scores for image and text similarity also tend to encourage copying and pasting the reference subject and follow prompts elsewhere. This is not the ideal integration of reference images and text prompts. Even worse, in rare cases, if half of the generated images have high identity preservation but low text alignment, and vice versa for the other half, we obtain no useful pictures with high information from reference images and good prompt fidelity, or otherwise poorly combined, but still have decent CLIP-I and CLIP-T averages. We tackle this problem in Sec. 4, where we propose a class of metrics named Similarity Weighted Diversity to evaluate how well the prompts and reference images are combined. When used with different pairs of metrics, new metrics can measure the quality of simultaneous reference and prompt similarity, which we refer as similarity-diversity balance.

We summarize **our contributions** as follows: **a)** We propose an anime dataset RetriBooru and the generalizable construction pipeline, which enables tracking and clustering of annotated outfits of the same character, and facilitates experimenting methods on referenced generation. **b)** Given the clothing annotation, we propose a new training pipeline for subject-driven generation by using a reference image different from the target but with the same identity, and verify its effectiveness by previous methods. **c)** We propose a new concept composition task, which learns to retrieve specific aspect information or various concepts from different reference images, and provide baseline results with our baseline architecture RetriNet. **d)** We propose a new class of metrics (SWD) to better evaluate referenced generation methods, explain its theoretical meaning, and demonstrate its effectiveness in evaluation. We conduct major experiments on RetriBooru using various methods and settings, drawing intriguing conclusions that also generalize to other domains with a larger scale.

## 2 RELATED WORK

**Reference architectures.** Initial approaches to inject information into the generation process apply to the attention blocks in the denoising U-Net. For example, in Prompt-to-Prompt (Hertz et al., 2022) and in Tune-A-Video (Wu et al., 2023), one can control the subject to ride a car instead of a bike while preserving the rest of the results. However, like in CustomDiffusion (Kumari et al., 2023), these methods are usually limited to more general concepts without as much geometric identity information as faces or clothes. IP-adapter (Ye et al., 2023) and FastComposer (Xiao et al., 2023), both aiming at subject face generation, do utilize image encoders, but with a shared feature vector for all blocks of the U-Net (as opposed to different vectors for different blocks in our RetriNet). MasaCtrl (Cao et al., 2023) adopts a parallel encoder with modified self-attention blocks so that the reference image is edited according to the new prompt. DreamTuner (Hua et al., 2023) tries to enhance referenced generation with an additional fine-tuning. StableVITON (Kim et al., 2023) adds body-related masks and segmentation for the virtual try-on application.

**Virtual try-on.** Starting from reproducing single objects (Chen et al., 2023; Li et al., 2023a), tasks of combining multiple objects or people have emerged (Liu et al., 2022; Xiao et al., 2023; Kumari et al., 2023; Ma et al., 2023), including virtual try-on, where given images of a person and an outfit, a generated image of the person wearing the clothes seamlessly is expected. It is more difficult than other composition tasks since there are more interactions between the body and clothes (Zhu et al., 2023; Li et al., 2023b; Institute for Intelligent Computing, 2023; Kim et al., 2023). A unique technical characteristic of virtual try-on is the fixed pose in the reference image as layout control, and only the appearance needs to be modified. For diffusion models, generating local appearances based on a given layout is much easier compared to generating geometry and appearance at the same time.

**Evaluation metrics.** Recent evaluation metrics are quite similar among the literature (Ruiz et al., 2023; Chen et al., 2023; Ruiz et al., 2023; Li et al., 2023a; Kumari et al., 2023; Ma et al., 2023), using CLIP (Radford et al., 2021) or DINO (Caron et al., 2021) features to measure similarity. CLIP-I is used to measure the similarity between the reference image and the generated image. CLIP-T is used to measure the similarity between the prompt and the generated image. Besides human user preference tests, (Achlioptas et al., 2023) aims to improve similarity scores. Nevertheless, those metrics offer simple trade-off between one another and fail to evaluate many more subtle aspects for referenced generation. In Sec. 4, we propose a new class of metrics to ameliorate this situation.

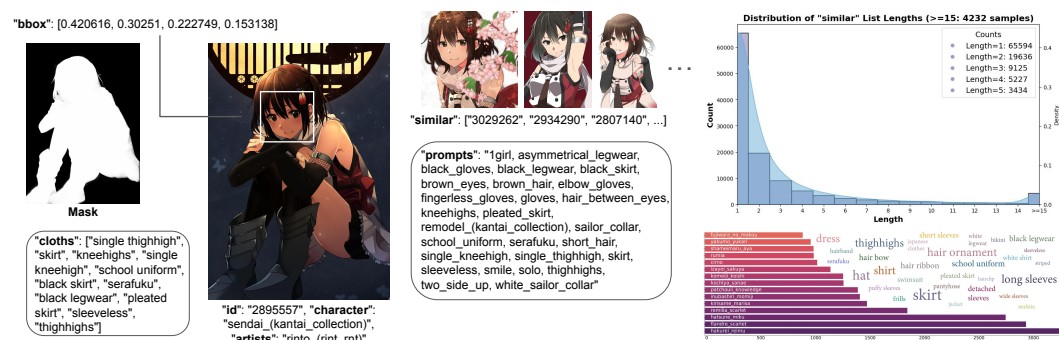

Figure 2: Details of RetriBooru dataset. **Left**: annotations of an individual sample. **Right**: distributions of lengths of the "similar" lists, top-15 characters with most appearances, and top-30 cloth tags.

# 3    RETRIBOORU DATASET

As discussed in Sec. 1, a concepts-labeled dataset is required to train a referenced generation model for retrieving given concepts interpretably. The motivation is to minimize the use of leaked information of the target, when training on the same image as both the reference and the target, which can form potential training shortcuts and cause overfitting. Existing datasets are single-tasked, lacking in size and types of identity, and may be harder to iterate on than anime figures (faces, for example). With the further annotated RetriBooru dataset, one can not only utilize cloth tags to constrain customization, apply masks and face boxes to refine details, but also leverage reference image(s) to serve as conditioning and train a more generalizable, robust condition encoder.

Table 1: Comparisons with datasets used in other literature.

| Dataset | Size | Category | Multi-images | Concept | Data source |
|---|---|---|---|---|---|
| DreamBooth (Ruiz et al., 2023) | ≤ 180 | Objects | ✓ | | Web |
| BLIP-Diffusion (Li et al., 2023a) | 129M | Objects | | | Mixture of datasets |
| FastComposer (Xiao et al., 2023) | 70K | Human Faces | | ✓ | FFHQ-wild (Karras et al., 2019) |
| CustomConcept101 (Kumari et al., 2023) | 642 | Objects and Faces | ✓ | ✓ | Web |
| RetriBooru (**Ours**) | 116K | Anime Figures | ✓ | ✓ | Danbooru (Branwen et al., 2020) |

It is very costly to collect multiple identities with multiple clothing images, and approaches that rely merely on CLIP features do not have enough sensitivity to reliably classify clothing identities. To ameliorate this, we leverage current large models to obtain clothing labels for our dataset with filtering heuristics. We construct our RetriBooru dataset from the Danbooru 2019 Figures dataset (Branwen et al., 2020), which cropped 855880 images of single anime figures from Danbooru by anime character detection model (Li & Shahjahan, 2018). We perform filtering by cloth tags, visual annotations using existing robust segmenter and detector, and clothing clustering using a visual-question answering (VQA) model. We use Instruct-BLIP (Liu et al., 2023) with `Vicuna-7B` as the VQA model, ask it descriptive questions for short answers, e.g., "list top two colors of the clothing", and cluster images of the same character wearing the same outfits using a strict heuristic that matches the output answers. Please see Supp. A.1 for processing details. In the end, RetriBooru retains 116729 training samples, where each training sample could serve as either the target image or a reference image for another target. Each of the reference image has the same character, outfits, and is drawn by the same artist as the target image. We show details of the dataset in Fig. 2. On the left is a sample image and its annotations, including basic information, mask and face bounding box, prompts and cloth tags, and a list of IDs which can access to other samples of the same identities. On the right are the distributions of the number of same-identity samples for each sample, top-15 characters present, and top-30 most frequent cloth tags. Each sample has at least one reference image in the "similar" entry, and there are 4232 samples which have ≥15 reference images. We also compare with proposed datasets in previous work in Tab. 1. Note that the larger the datasets are, the less functionalities they enable due to annotation costs, and vice versa. In comparison, RetriBooru hits the best balance between size and capabilities, where there are multi-images of an identity (similar images to the target), as well as concept-aware labeling (two concepts per sample, face and cloths). CustomConcept101 (Kumari et al., 2023) is the most similar dataset to ours serving the same functionalities, but only contains image clusters of 101 realistic objects, in total 642 images. Using automated pipeline and strict answer matching to group multiple images by identities, RetriBooru is able to collect accurate annotations, and enable learning to compose separated concepts which will be further discussed.

(a) Prompt 1          (b) Prompt 2

Figure 3: Qualitative results of **IP-Adapter** trained on RetriBooru. We choose two image-prompt pairs and compare with different scales. Each row has the same scale and each column has the same seed. Our **-b** pipeline provides better balanced results given a fixed scale, and keeps fusing image and text conditioning at various scales, outputting good generation even when **-a** scale is off.

## 3.1 BENCHMARKING EXISTING MODELS

**Training.** We adopt FastComposer (Xiao et al., 2023), ControlNet (Zhang et al., 2023), IP-Adapter (Ye et al., 2023), and Kandinsky (Razzhigaev et al., 2023) to finetune on our dataset (initialized with `SD v1.5` weights), as they are the latest open source methods that process conditional images with distinct designs: FastComposer injects identifiers to help place objects (in conditional images) into one image; ControlNet learns pixel-wise control and exploits spatial structures on purpose, passing conditioning to the U-Net via zero convolutions; IP-Adapter passes reference images to a frozen encoder and additionally learn to adapt the features, adding the same outputs to all layers in U-Net; Kandinsky adopts a prior network (taking prompts and reference) and U-Net (target and reference) that can be pre-trained separately. Due to the limited computing power, we finetune with resolution 256 and FP16 for all methods without rigorous hyper-parameter searching, using 8 NVIDIA V100 GPUs. Note that we do not pursue the peak performance but conduct sufficient training for our analysis, and we do not leverage extra training tricks such as additional loss updates on face parts. To verify our motivation and assumptions, we develop two training settings: **-a** denotes existing training pipeline, which applies two different transforms the same image as target and reference (conditional) image, and **-b** adopts a same-identity but different image from target as the reference image, everything else unchanged. It would be more fair and informative to compare two settings within the same method to better evaluate the impact of target information leakage.

**Evaluation.** Our evaluation set consists of 40 in-domain anime characters and 10 out-domain characters present in recent anime series, as well as 5 prompts from in-domain characters. For each character-prompt pair, we generate images with 384 resolution with 5 random seeds and 4 samples per seeds. In total, we have 5000 generated images for evaluation for each model. We use CLIP scores between generated images and reference image, generated images and text prompt to evaluate identity preservation and text alignment, denoted as CLIP-I and CLIP-T. We also multiply two quantities element-wisely to roughly evaluate the balance. See Supp. A.3 for detailed training and evaluation settings, including fixed parameter choices for the models.

**Results analysis.** We report CLIP scores in Tab. 2 with a scatter plot to compare model balances between image and text conditioning. As **-b** results suggest, using a different image as the reference image in general improves image-text balance across diverse methods, motivating a better fusion by reducing the target information leakage. Among the finetuned models, `Kandinsky-0.1-b` achieves the best identity preservation, `FastComposer-0.5-b` achieves the best text alignment, and `IP-Adapter-0.5-b` achieves the best balance. ControlNet models tend to fit to text and

Table 2: Comparison of models by CLIP-I and CLIP-T scores. **Left**: average scores across validation prompts. **Right**: A scatter plot that discloses the balance between text and image prompt alignment. Overall, `IP-Adapter-0.5-b` achieves the best balance.

| Model | CLIP-I↑ | CLIP-T↑ | CLIP-I·CLIP-T |
|---|---|---|---|
| FastComposer-0.5-a | 0.6406 | 0.2616 | 0.1677 |
| FastComposer-0.5-b | 0.6416 | **0.2637** | 0.1693 |
| ControlNet-a | 0.7476 | 0.1787 | 0.1339 |
| ControlNet-b | 0.6384 | 0.2594 | 0.1658 |
| Kandinsky-0.2-a | 0.6652 | 0.2259 | 0.1503 |
| Kandinsky-0.2-b | 0.7272 | 0.2093 | 0.1523 |
| Kandinsky-0.1-a | 0.7400 | 0.2028 | 0.1502 |
| Kandinsky-0.1-b | **0.8019** | 0.1941 | 0.1557 |
| IP-Adapter-0.5-a | 0.7692 | 0.2064 | 0.1586 |
| IP-Adapter-0.5-b | 0.7283 | 0.2382 | **0.1731** |

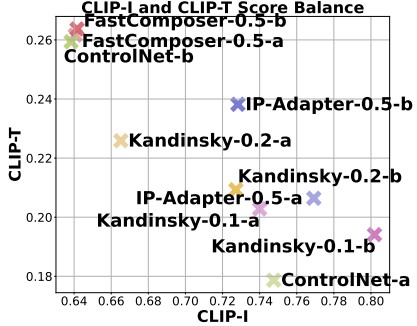

produce a worse identity preservation under **-b**. This is due to its zero convolutions' pixel-to-pixel control, which essentially cannot support **-b** setting. It makes learning pixel-wise mapping from a different reference to the target more challenging than overfitting to the texts in search of convergence. This also indicates why later methods (Ye et al., 2023; Li et al., 2023a), including our RetriNet, adopt cross attentions to connect conditional encoder to the U-Net for flexible control. To qualitatively demonstrate the benefits of our **-b** setting, we choose two image-prompt pairs to visualize inference results of IP-Adapter for $\texttt{scale} = [0.25, 0.5, 0.75]$ in Fig. 3. We observe that: a) When given a fixed scale, our **-b** combines reference image and prompt better than original **-a**. b) Given various scales, **-b** continues to produce better balanced generation, showing robustness against over-fitting, saving time for searching inference parameters to generate balanced outputs. c) When **-a** collapses with off scales (0.75), **-b** can still generate plausible, better results. These further validate the effectiveness of our proposed new training scheme.

Overall, RetriBooru is superior with increased sample size, extra identity annotations, multi-concepts functionality (Tab. 1), and more performance friendly. With identity annotations, we can not only enable a better training pipeline using different reference images, but also enable future tasks which learns to retrieve different concepts accurately from different reference images.

## 4 SIMILARITY WEIGHTED DIVERSITY

Although our new training pipeline enabled by annotating clothes has been justified, there lies a discrepancy between qualitative results and metric scores. The goal of subject-driven generation is to generate images flexibly while maintaining subject identity (object or personal) and combining text prompts well. As discussed in Sec. 2, existing metrics measure image and text similarities separately. However, observe that the better qualitative results of **-b** setting are only roughly reflected in Tab. 2, and decoupled averages of the metrics are not good measures of qualities of subject driven images. Oftentimes, some generated images tend to resemble reference images while others follow texts, especially when generation parameters are not well-optimized, yielding inflated averaged scores. We provide a class of metrics to combine similarity and diversity metrics, which require both extracting the desired information from reference images and combining reference information with texts. Hence, it is better suited for referenced generation. Our metrics also benefit from continued improvements in the precision of similarity and diversity measurements. For instance, CLIP-I is not yet an optimal measurement identity preservation because preserving other irrelevant style, layout and semantic information also contributes to CLIP-I. In practice, these irrelevant information usually interfere with text prompts, resulting in artifacts.

A reference image generation model can be written as a function $f : r, \theta_r \to g$, where $r$ represents reference image(s), $\theta$ is the collection of all generation parameters for the run (the prompt, scheduler, guidance scale etc.), and g is the generated image. We denote div and sim (diversity, similarity) measurement functions as $\text{div} : r, g \to \mathbb{R}$, $\text{sim} : r, g \to \mathbb{R}$. Under this notation, for a pair of measurement functions div and sim, we define:

$$\text{SWD}(\text{div}, \text{sim})(f) = \frac{1}{|(r, \theta_r)|} \sum_{r,g=f(r,\theta_r)} \text{div}(r, g) \cdot \text{sim}(r, g) \tag{1}$$

where each tuple $(r, \theta_r)$ represents a test generation with the reference images $r$ and parameters $\theta_r$. $|(r, \theta_r)|$ is the total number of such runs. We illustrate with two concrete examples.

**Measuring explicit diversity.** As briefly mentioned, in (Wang et al., 2024b), authors noticed that some approaches can inadvertently impact text control. A notable example is the inability to seamlessly integrate the facial area with the background style. This limitation highlights a trade-off between face fidelity and text control. This trade-off of face fidelity and text fidelity could be masked by simply considering CLIP-I and CLIP-T separately. A worse trade-off of having high CLIP-I, low CLIP-T for half of the images and low CLIP-I, high CLIP-T for the other half results in medium CLIP-I and CLIP-T scores, but generates very low amount of ideal images where identity is preserved, text control is followed, and image-text is well fused. Since text control explicitly dictates diversity from input reference images, we take CLIP-I to be sim, CLIP-T to be div and can consider this as a better measurement for explicit (prompt-driven) diversity. In this case, SWD(CLIP-T, CLIP-I) works as a filter for images that are both consistent to the reference images and follows the text prompts. Because of Cauchy-Schwartz inequality:

$$\sum_{r,g=f(r,\theta_r)} \text{div}(r, g) \cdot \text{sim}(r, g))^2 \leq \sum_{r,g=f(r,\theta_r)} \text{div}(r, g)^2 \cdot \sum_{r,g=f(r,\theta_r)} \text{sim}(r, g)^2 \tag{2}$$

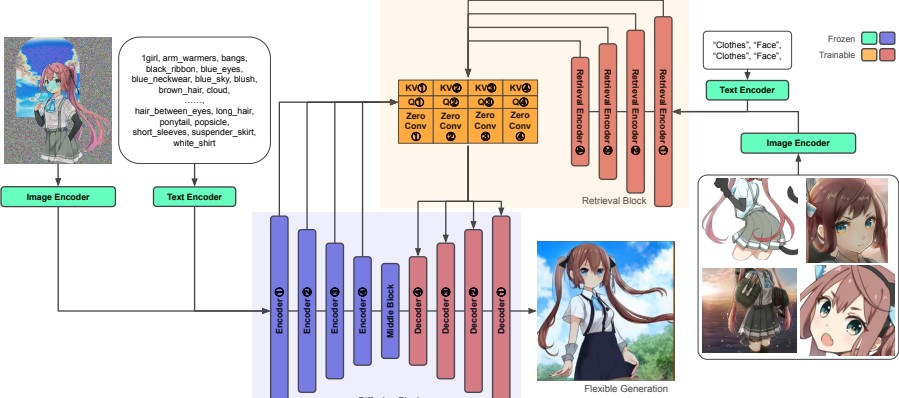

Figure 4: An illustration of concept composition task on RetriBooru using RetriNet. We pass reference concepts into the retrieval encoder to encode precisely only the relevant information. We pass noisy target image and prompt made with Danbooru tags to a copy of denoising U-Net. Note that we process reference images, texts, and time with corresponding frozen encoders, and we have frozen encoder and middle blocks of SD. We designate the embeddings derived from target images and prompts as Query (Q), while the embeddings from reference images and text serve as Key and Values (KV). Following the cross attention layers are zero-convolution layers.

The metric encourages simultaneous achievement of reference consistency and prompt consistency, i.e. better distribution of CLIP-T with respect to CLIP-I.

**Measuring implicit diversity.** When generating faces, different head poses and expressions are implicit to the distribution. When generating anime characters, the characters could be in different poses and expression too. These are implicit data distribution diversities. Then SWD (as we will define for anime images) captures the fact that generated images should preserve the identities but still not be rigid and have a fixed pose. We can measure this by asking a VQA system (InstructBLIP (Liu et al., 2023) `Vicuna-7B`) descriptive questions about explicit attributes and compare answers, and we denote this metric as the VQA score. For instance, we can ask the VQA model about head orientation, facial expressions, body gestures, activities and so on. To compare two generated images, we embed the answers into vectors and compute the cosine similarity. See more details in Supp. A.2.

We will use SWD(VQA, CLIP-I), SWD(VQA, CLIP-T), and SWD(CLIP-I, CLIP-T) to evaluate Retribooru tasks in Tab. 3. We have also seen one simultaneous independent work using face distance as sim and LPIPS as div in (Wang et al., 2024a) (Trusted Div. in Sec. 4.1).

## 5 CONCEPT COMPOSITION TASK

We extend our **-b** setting in Sec. 3.1, using multiple reference images from the "similar" entry of the target. In the following sections, we refer to directly using various reference images as the "reconstruction" task, given its unchanged training goal despite various reference. We further propose a concept composition task, where different reference images carry different concepts for the condition encoder to retrieve and compose. As shown in Fig. 1, we use annotated bounding boxes and masks to separate desired concepts from the reference images, and pass to the condition encoder along with the corresponding texts. Note that we randomly choose $N$ reference images and obtain either "face" or "clothes" concepts independently, instead of building $N$ reference images by dividing each of $N/2$ images into two concepts. In rare cases, we replace the "clothes" reference with its whole image, denoted as "figure" reference, if the face bounding box occupies the majority of the image. See Supp. A.4 for implementation details. Likewise, for reconstruction task, we directly pick $N$ reference images and pass them to the condition encoder with texts "figure". Concept composition is a more difficult task than reconstruction, as it needs to not only recognize the concepts in the reference, but also learn to compose them properly.

**Expectation for $N$.** Ideally, We want $N$ to be as large as it can be supported by sufficient annotations. When the reference image is the same as the target, the conditional encoder cannot constrain what information is passed and learned, forming shortcuts based on inductive bias such as geometric structures, shapes and gestures, leading to overfitting. Hence, information from inductive bias is leaked to the model for convergence, while the characteristics of an identity is insufficiently learned. However, when the reference image maintains only the identity but has other attributes differently

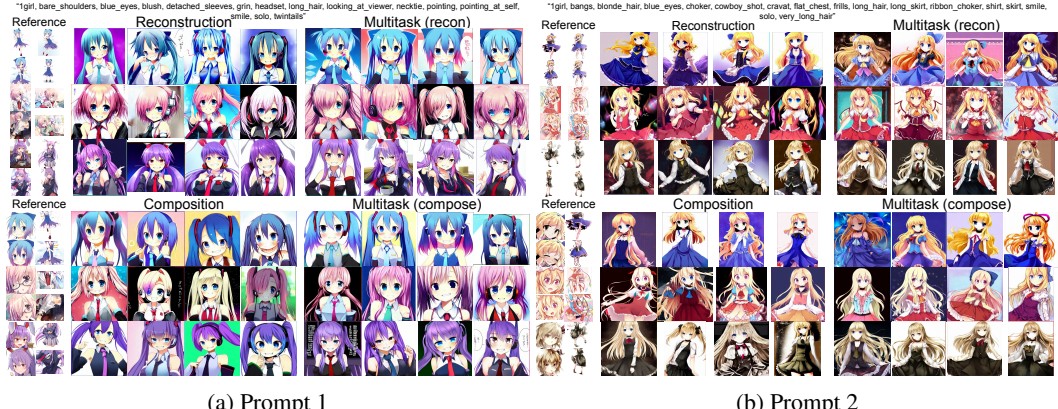

(a) Prompt 1             (b) Prompt 2

Figure 5: Qualitative results on the RetriBooru dataset. We use RetriNet for reconstruction, concept composition, and a multitask setting.

such as orientations, poses, activities, it naturally breaks shortcuts based on the inductive bias, limits the model to learn identity from only characteristic features, and thus learns from the correct and trustworthy information. With the number of annotated reference images increasing, it better approximates the data distribution with more discrete, diverse samples of the same identity, and applies more constraints to calibrate on retrieving the correct information. In the best but unrealistic case, "similar" represents the identity's space.

## 5.1 RETRINET

Observed in Tab. 2, ControlNet-**b** (Zhang et al., 2023) overfits to text prompt to avoid the hard task of learning image conditioning. However, we can adopt cross attention layers to connect control module to the U-Net and thus enabling flexible control. This approach can also support training on our new task. We propose a modified architecture RetriNet, which retrieves concepts from references via a retrieval encoder and inject them into the denoising U-Net via cross attention layers, without relying on geometric controls. *To the best of our knowledge, we are the first to show this structure works without the need for geometric information or layout guidance, such as key-points and semantic masks.* Fig. 4 demonstrates our architecture design and the pipeline for training concept composition task on RetriBooru. This approach unifies geometric retrieval instead of pre-processing for different key-points and masks for each different task. When geometric or layout information are given (as some of the simultaneous and independent virtual try-on and video generation methods we surveyed have done), the abilities demonstrated are the encoding of reference appearances. We push the abilities even further, showing that RetriNet can simultaneously generate and reconstruct geometry and appearance. We also show that RetriNet can retrieve and compose geometry and appearance. See Supp. A.4 for more details. Note that unlike concurrent work such as IP-Adapter (Ye et al., 2023) that adds a common cross attention output to U-Net layers, RetriNet computes cross attentions and adds to U-Net decoder for each layer individually, as Fig. 4 marks the corresponding layer numbers.

**Training objective.** Let $\mathbf{M}_{\text{tgt}}$ be the mask of the valid area of the target image (excluding padding). Let $\mathbf{M}_{\text{face}}$ be face masks for each target. Denote the original reconstruction loss as $\mathcal{L}(\epsilon, \epsilon_\theta(\cdot))$, our loss between the target and reconstruction then becomes

$$\mathcal{L}' = \mathcal{L}(\epsilon \mathbf{M}_{\text{tgt}}, \epsilon_\theta(\cdot)\mathbf{M}_{\text{tgt}}) + \lambda \mathcal{L}(\epsilon \mathbf{M}_{\text{face}}, \epsilon_\theta(\cdot)\mathbf{M}_{\text{face}}). \tag{3}$$

## 5.2 EXPERIMENTS

**Setup.** We conduct both concept composition and reconstruction tasks on RetriBooru with RetriNet, providing baseline results for future research. We also combine reconstruction and concept composition as one retrieval task, denoted as multitask. We follow the standard denoising training. SD encoder, decoder, and retrieval encoder blocks are initialized with corresponding blocks in SD v1.5 weights. We train on $8\times$ NVIDIA V100 GPUs, with resolution 384, FP32, batch size 1 but accumulate every 8 batches. Training 9000 steps roughly requires 72 GPU hours. We perform one longer 45000-step training for each task, with trainable retrieval block and U-Net decoder, and a $p = 0.5$ drop-rate to drop text prompt to the U-Net. By default, we use $N = 4$ reference images. We also conduct various ablation studies for 9000 steps and run inference outputs at resolution 384, with the same evaluation set and procedure specified in Sec. 3.1, except that we prepare reference images the same way as training our tasks. See Supp. A.4 for other fixed choices of hyper-parameters.

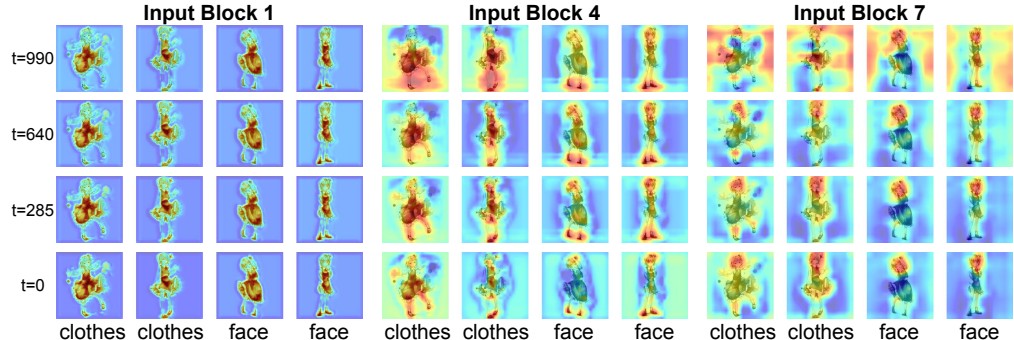

Figure 6: Attention maps of early to deeper layers in the retrieval block.

**Quantitative analysis and ablations.** We record quantitative results in Tab. 3, and visualize balances between diversity (text) and similarity (image) in the scatter plots on the right. "R-, C-, M-" represents reconstruction, concept composition, and multitask, followed by number of training steps. "P-" denotes a prompt droprate different from the default 0.5. "LockD, LockB" refer to freezing the U-Net decoder and freezing both the U-Net decoder and retrieval encoder. We summarize that: **a)** Training w/o locking the frozen image encoders adopted in (Ye et al., 2023; Li et al., 2023a) results in better generalization. Observe that freezing either network module harms the performance for reconstruction and concept composition, with "-LockB" having more impact. This is due to the extra objective of the retrieval encoder to recognize the information to retrieve from the conditioning, and it can not adopt a frozen image encoder and decoder without such training. **b)** A lower drop rate helps guide the model learning by more presence of text prompts for the more difficult concept composition, where retrieving conditioning from reference images is more challenging than reconstruction task. This explains why C9kP025 achieves best similarity-weighted diversity (SWD) scores among concept composition models. Given different tasks of different difficulties, models are more sensitive to the prompt drop-rate. We argue that C9kP025 tends to fit to text prompts during training in pursuit of faster convergence. While imbalanced convergence is not encouraged, this finding suggests future work on constraining the existing objectives. **c)** As the top scatter plot suggests, longer training benefits CLIP-I and CLIP-T balance, despite that longer training tends to trade CLIP-T for improved CLIP-I. **d)** -45k do not necessarily outperform -9k models. This indicates that the training on RetriBooru can converge fast at earlier phase. Combining with **c**, we hypothesize that later training mainly aims at better fusing the image reference and the text prompt.

**Qualitative analysis and visualizations.** We provide qualitative results from -45k models of three tasks on RetriBooru dataset and record generated images in Fig. 5, selecting three characters for each of the two prompts. "Multitask (recon)" denotes running image generation in the same fashion as reconstruction task, and vice versa. We observe that R45k has achieved best image quality and best diversity-similarity balance among three models: light-reflecting hair, less artifacts, diverse details, etc. This also aligns with our SWD scores, where R45k achieves best VQA_CLIP-T and CLIP-I_CLIP-T among the three. M45k in general has the second best image quality, but sometimes

Table 3: Baseline results on RetriBooru. **Left**: Average scores across validation prompts, and we mark the highest for each task. **Right**: A scatter plot that discloses the balance between diversity (text) and similarity (image). Longer training achieves better CLIP-I CLIP-T balance.

| Model | VQA↑ | CLIP-I↑ | CLIP-T↑ | VQA_CLIP-I↑ | VQA_CLIP-T↑ | CLIP-I_CLIP-T |
|---|---|---|---|---|---|---|
| R9kLockD | 0.4276 | 0.7022 | 0.2120 | 0.2985 | 0.0909 | 0.1487 |
| R9kLockB | **0.4383** | 0.6856 | 0.2077 | 0.2992 | 0.0912 | 0.1422 |
| R9kP025 | 0.4234 | 0.7139 | 0.2306 | 0.2998 | 0.0988 | 0.1636 |
| R9kP075 | 0.4012 | **0.7384** | 0.2037 | 0.2943 | 0.0823 | 0.1500 |
| R9k | 0.4372 | 0.7008 | **0.2331** | **0.3044** | **0.1027** | 0.1628 |
| R45k | 0.4161 | 0.7328 | 0.2311 | 0.3029 | 0.0970 | **0.1687** |
| C9kLockD | 0.4276 | 0.7002 | 0.2138 | 0.2979 | 0.0917 | 0.1495 |
| C9kLockB | 0.4297 | 0.6979 | 0.2113 | 0.2984 | 0.0910 | 0.1473 |
| C9kP025 | **0.4390** | 0.6941 | **0.2377** | **0.3031** | **0.1050** | **0.1645** |
| C9kP075 | 0.4280 | 0.7007 | 0.2230 | 0.2980 | 0.0963 | 0.1557 |
| C9k | 0.4348 | 0.6976 | 0.2318 | 0.3016 | 0.1017 | 0.1612 |
| C45k | 0.4251 | **0.7111** | 0.2315 | 0.3004 | 0.0994 | 0.1640 |
| M9k | **0.4377** | 0.6963 | **0.2334** | **0.3031** | **0.1029** | 0.1621 |
| M45k | 0.4142 | **0.7221** | 0.2259 | 0.2973 | 0.0945 | **0.1625** |

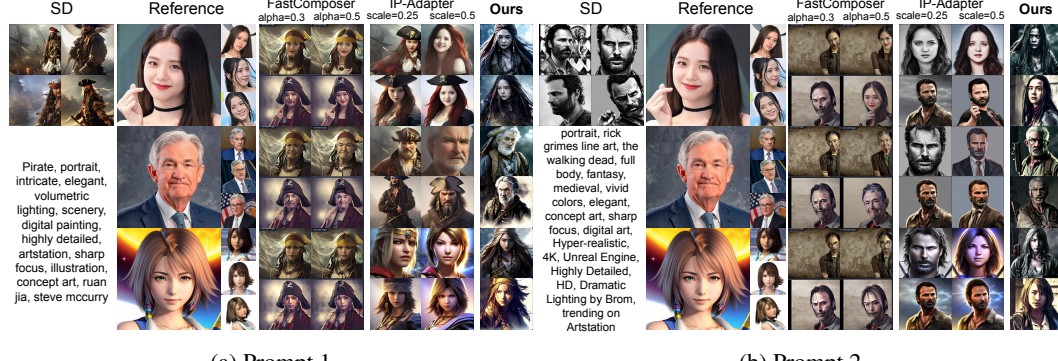

(a) Prompt 1                                  (b) Prompt 2

Figure 7: Qualitative comparisons between RetriNet and SOTA methods on human faces.

retain the background and accessory details of the reference images, affecting diversity. C45k displays the most flexible and diverse generation, and follows well the text prompt. However, since concept composition is more challenging, we can observe a few more artifacts such as round head and limb issues. These observations are also reflected by these models' SWD scores. This further validates the effectiveness of evaluating diversity-similarity balance using the proposed VQA score and SWD metrics. Additionally, we visualize attention maps at different layers of the retrieval encoder from the C45k model in Fig. 6. Given a simple background, the retrieval encoder is able to locate the identity at an early stage. With deeper layers, the identifying texts are able to highlight the corresponding semantics on the reference images, and the attention regions are further refined with timesteps. At $t = 0$ for Input Block 7, "clothes" and "face" can be properly highlighted as intended. Note that the "face" region can sometimes also be highlighted by "clothes". It is due to the additional loss update on the face bounding box we adopt per Eqn. 3, as well as the cropping for "face" reference images for our training.

**Generalizing to other domain.** Our observations and conclusions from both experiment sections also generalize to other domain. Specifically, adopting different reference image(s) with the same identity as target (**-b** setting), unfreezing conditional encoder and U-Net decoder for the new pipeline, and applying a text droprate contribute to a better subject-driven generation model. We follow these conclusions to pre-train RetriNet on existing human face datasets (Rothe et al., 2015; Liu et al., 2018; An et al., 2021), and provide qualitative comparisons with public models of FastComposer and IP-Adapter for faces in Fig. 7. Baseline generation results from vanilla SD are also included. Our training pipeline keeps a better identity-diversity balance and combines prompts and reference images better, while other methods maintain a worse trade-off. The ability to generalize to more complex data also depends on the computation resources, where higher resolution accommodates more details. Our largest possible training resolution is 384, limited by V100. Note that we do not search for the optimal parameters for human face training as it is not our focus; this experiment only serves to validate that our proposed dataset and training settings can provide valuable observations and generalize to other data domains.

## 6 CONCLUSION

In this paper, we propose a new dataset RetriBooru, which enables new training pipelines and tasks with extra identity annotations. We successfully show that using same-identity pairs (reference and target image have the same identity but are not the same image) results in more robust generalizations, and also enable training for composing different concepts from different reference images, using a newly designed model. Moreover, we propose a new group of metrics to weight similarity (image identity) and diversity (text prompt), which aligns well with visualizations and provide better evaluation. Our dataset facilitate research with constrained computes and provide enough samples to yield meaningful analysis that can generalize. Our study sheds light on improving current training procedure and objectives.

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
