# A SUPPLEMENTARY

In this section, we provide supplementary materials to the main paper, including further details, limitations, and maintenance plans. All coding and annotation files will be released. Again, RetriBooru is an anime dataset based on the existing open-source Danbooru 19 Figures dataset (Branwen et al., 2020). Our main contribution lies in the extra annotations, including labeling clothing identities, which enable new training pipeline and tasks as mentioned in the main paper.

## A.1 DATA COLLECTION AND PROCESSING

**Filtering.** We first clean and filter noisy samples by unwanted tags such as "monochrome" and "sketch" images. We also remove images with multiple character tags. In order to cluster cloth tags in the following steps, we scrape clothing tags from Danbooru, and filter trivial tags from them such as tags for shoes and accessories. We then filter samples which do not contain any of the remaining 1298 cloth tags for further clustering. At the end of the filtering stage, we have obtained 599192 images, each with a single character and artist, and contains at least one meaningful cloth tag. **Note that all images are safe-for-work** as the original Danbooru 19 Figures (Branwen et al., 2020) has performed this filtering.

**Vision annotations.** We create segmentation masks of characters by IS-Net (Qin et al., 2022), as well as head bounding boxes by YOLO-v5 (Jocher et al., 2022). **Both models were pre-trained on the largest Danbooru 2021 dataset (Anonymous et al., 2022) to secure inference quality.** Segmentation masks and head bounding boxes allow us to separate different concepts of an anime character into whole figure, clothes, and face, which facilitate future training of choice. Moreover, both annotations help create masks for these concepts to provide refined training, such as extra weighted loss computation on the masked parts.

**Clustering clothes.** Labeling clothing identity is a difficult yet crucial process in order to train with consistent samples of a concept class. We use Instruct-BLIP (Liu et al., 2023) with `Vicuna-7B` and a heuristic to cluster answers. Instruct-BLIP is a visual question-answering (VQA) model which takes texts and images jointly and outputs understandable answers for further processing.

We first group images by the same character and artist to align the artistic styles. For $N_c$ images of each character-artist class, we ask the VQA model to "List top two colors of the character's cloths". We then iterate these samples pairwise to cluster them based on matching answers in $\mathcal{O}(N_c^2)$. Note that it is a strict heuristic and the order matters, i.e., "black and pink" and "pink and black" reflect different probability distributions and will not be clustered together. In the end, we obtain 116729 samples, where each image sample is now connected to other images with the same character, artist, and the clothing. Algorithm 1 summarizes our approach in a Python-like pseudo-code, which can generalize to future data construction.

There are two **limitations** to this approach. First, the proposed question has overlooked finer details in pursuit of stable outputs, such as textures and design patterns of the clothes. Second, in order to filter out unrelated answers, the matching heuristic has rejected 80% of initial data strictly, including correct and usable samples. The more detailed and abstract the questions are, the more irrelevant and inaccurate responses can be sampled. With more advanced VQA models in the future, more questions can be better understood and our clustering performance can be further improved.

---

**Algorithm 1:** Cluster Images by VQA answers

---

**input** : A list of images $X$ with length $N$; a list of $M$ questions $Q$
**output**: A dictionary $D$ with length $N$

$D_0 \leftarrow \text{Initialize}(X), D \leftarrow \text{NewDict}$ /* $D_0$:K character-artist pairs as keys */
**for** $X_k \in D_0 . \text{values}()$ **do**
    $\text{Similar}_k \leftarrow [] * \text{len}(X_k)$;    /* Pair-wisely compare answers */
    **for** $i, j \leftarrow 0$ **to** $\text{len}(X_k) - 1$ **and** $i \neq j$ **do**
        $[\text{ans}_i] \leftarrow \text{Decode}(\text{InstructBlip}(X_k[i], Q))$;
        $[\text{ans}_j] \leftarrow \text{Decode}(\text{InstructBlip}(X_k[j], Q))$;
        **if** $[\text{ans}_i] == [\text{ans}_j]$ **then**
            $\text{Similar}_k[i] . \text{append}(X_k[j])$
        **end**
    **end**
    **for** $i \leftarrow 0$ **to** $\text{len}(X_k) - 1$ **do**
        $D[X_k[i]] \leftarrow \text{Similar}_k[i]$
    **end**
**end**
**return** $D$

---

## A.2 MORE DETAILS OF THE VQA SCORE

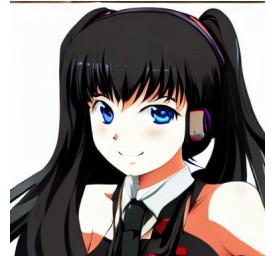

What are the facial expressions, angles, poses, body gestures and the current actions of the character?

The character in the image is displaying a friendly and approachable expression with a slight smile. The eyes are large and bright, indicating attentiveness or excitement. The character's head is tilted slightly forward, which can be a sign of interest or engagement. Her body is facing the viewer directly, suggesting openness. There are no significant body gestures as only the head and shoulders are visible. The character is not engaged in any discernible action other than posing for the image.

What are the facial expressions and hairstyles of the character?

The character in the image has a gentle smile with a hint of mischief, highlighted by bright blue eyes. The hairstyle is long, straight, and black with bangs neatly cut across the forehead. There are also what appear to be two small red clips or ties accenting the sides of the hair, which is a common accessory in anime-style character designs. The ears are adorned with what seems to be a set of headphones, adding to a modern, youthful look.

What are the poses, current actions and the clothing of the character?

The character in the image is depicted in a frontal pose with a slight tilt of the head to the right, giving a gentle, engaging look towards the viewer. The character's current action seems to be a subtle smile with a finger placed near the lips in a thoughtful or playful gesture. The clothing appears to be a school uniform, consisting of a white blouse with a black trim collar, complemented by a red necktie. The character also has long black hair and blue eyes, and is wearing what seems to be a set of headphones around the neck.

Generated (Tokitou Muichirou + P1)

Figure 8: Demonstration (also motivation) of VQA-based evaluation using GPT-4 on a generated image. An optimal VQA model can function as human evaluation and provide detailed, understandable descriptions. Using text embedding from accurate answers, we can properly evaluate the details of the image beyond semantics and measure the diversity based on distance metrics. Moreover, we can adjust prompt questions to focus on different aspects as shown.

We use a VQA model to measure $\text{div}(r, g)$, as Figure 8 gives an optimal example of the VQA evaluation. Given a set of images generated by the same reference images and prompt, we first ask the VQA model a few descriptive questions focusing on different aspects for each image. We then convert the answers to text embedding using the CLIP model, and compute the pairwise cosine distance between the embedding of the generated image and that of the reference image. In practice, we use InstructBLIP (Liu et al., 2023) with `Vicuna-7B` as our VQA model, and ask the second and third questions as shown in Figure 8. We also include a few example answers in Table 4, which are human-understandable and informative enough for the following evaluation.

The motivation of the proposed VQA score is to evaluate image details that are overlooked by CLIP score. While the CLIP score focuses on geometry and semantics, our VQA score evaluates generated details which are under-conditioned, or controlled by text prompts, such as facial expressions, gestures, and clothing details. These perspectives also demonstrate the flexibility of generation. Our VQA score hence can evaluate the diversity of the generated images, which contributes to an SWD metric.

Table 4: Example answers from InstructBLIP with `Vicuna-7B`, based on images generated by various prompts and references, using the pre-trained RetriNet on the concept composition task.

| Q1 | Q2 |
|---|---|
| The character in the image is wearing headphones and has a ponytail hairstyle. Her facial expression appears to be cheerful or happy, as she is smiling and giving a thumbs-up gesture. | The character in the image is a young girl with long blonde hair, wearing headphones and a tie. She appears to be smiling or posing happily while listening to music through her headphones. Her clothing consists of a white shirt, black tie, and pigtails. |
| The facial expressions and hairstyles of the character are cute, innocent, and adorable. She has a big smile on her face and is wearing a sailor-style outfit with blue eyeshadow. Her hair is short and styled in a bobbed or ponytail style. Additionally, she appears to be posing seductively for the camera. | The character in the image is a blue-haired anime girl wearing a swimsuit. She is posing and leaning forward, with her hands on her hips. Her clothing consists of a one-piece swimsuit that exposes her midriff. Additionally, she appears to be standing on top of a wooden table or bench. |
| The character in the image is wearing a black hat and has white hair. Her facial expression appears to be neutral or calm, while her hairstyle is long and flowing down her back. | The character in the image is an anime girl wearing a black hat, white dress, and brown boots. She is sitting on a bench with her legs crossed while holding a red strawberry between her toes. Additionally, she appears to be smiling at the viewer. This combination of poses and clothing creates a unique and eye-catching image that showcases the character's personality and playful nature. |
| The character in the image is wearing a black dress and has long blonde hair. Her facial expression appears to be serious or thoughtful, while her hairstyle is styled with bangs that frame her face. | The character in the image is a young girl with long blonde hair, wearing a black dress. She has her arms crossed over her chest and appears to be posing or holding herself in a particular way. Her clothing suggests that she may be dressed up for a special occasion or attending a formal event. In terms of current actions, there is no specific action depicted in the image, as it is an illustration rather than a live-action scene. |
| The character in the image has a pink hairstyle and is wearing an angel costume. Her facial expression appears to be neutral, with no visible emotions or expressions on her face. Additionally, she has large wings behind her back, which further emphasizes her role as an angel-like character. | The character in the image is a pink-haired anime girl wearing a blue outfit with wings on her back. Her poses and current actions suggest that she is flying or hovering above the ground, possibly using her wings to propel herself through the air. She appears to be graceful and agile while maintaining control of her flight. |

## A.3 More Details of the Benchmarking Training

To benchmark selected methods in RetiBooru, we adopt official code bases for FastComposer (Xiao et al., 2023) and IP-Adapter (Ye et al., 2023), and training scripts from the HuggingFace `diffusers` library for ControlNet (Zhang et al., 2023) and Kandinsky (Razzhigaev et al., 2023). For all methods, we load pre-trained weights for SD U-Net from `runwayml/stable-diffusion-v1-5` on HuggingFace, and `openai/clip-vit-large-patch14` as the image encoder. We train with the same resolution 256, batch size 1, gradient accumulation steps 8, learning rate $1^{-5}$, and precision FP16, on 8 NVIDIA V100 GPUs. We pre-train each model (each method, each setting) for 24 GPU hours. Inference parameters are in general kept default, and we adopt guidance scale 7. We keep other default settings as specified in their official code bases, with the following unique specifications and modifications of each method:

**FastComposer:** We add the default identifier "A $< |$image$| >$" to the start of our prompts, and set `uncondition_prob` to 0.5 to drop text prompts. We set the number of objects in the image to 1 and we only have one `object_types` which we by default choose "person". We use our masks to obtain object images and set the object resolution to 256. For **-b** training pipeline, we choose a random image from the "similar" entry of our target annotation and process it as the object image, which will be passed as the conditioning. During inference, we adopt the default $\alpha = 0.7$.

**IP-Adapter:** We prepare our text prompts and images by the default settings, and follow its default logics to drop texts and conditional images with small probabilities. For **-b** setting, we choose a random reference image as the conditioning, which is referred as `clip_image` in its code base. During inference, we choose `scale` $= 0.5$ for both qualitative and quantitative evaluation, and we also infer with $[0.25, 0.75]$ for a more comprehensive visual comparisons.

**ControlNet:** Similar to IP-Adapter, we keep the default settings from the `diffusers` library except that we set `proportion_empty_prompts` to 0. For **-b** setting, we choose a random reference image as the conditioning image (`conditioning_pixel_values`). Inference parameters are kept default.

**Kandinsky:** We pre-train prior network and decoder network separately, using the same set of hyper-parameters. Like training ControlNet, we we set `proportion_empty_prompts` to 0. For **-b** setting, we choose a random reference image. During inference, we infer with `strength` $= [0.1, 0.2]$ as the paper suggests smaller strengths.

Figure 9 provides expanded scatter plots to evaluate the similarity-diversity balance, clustering all 5000 inferred samples for each model. **-b** settings (oranges) help improve the balance as we observe

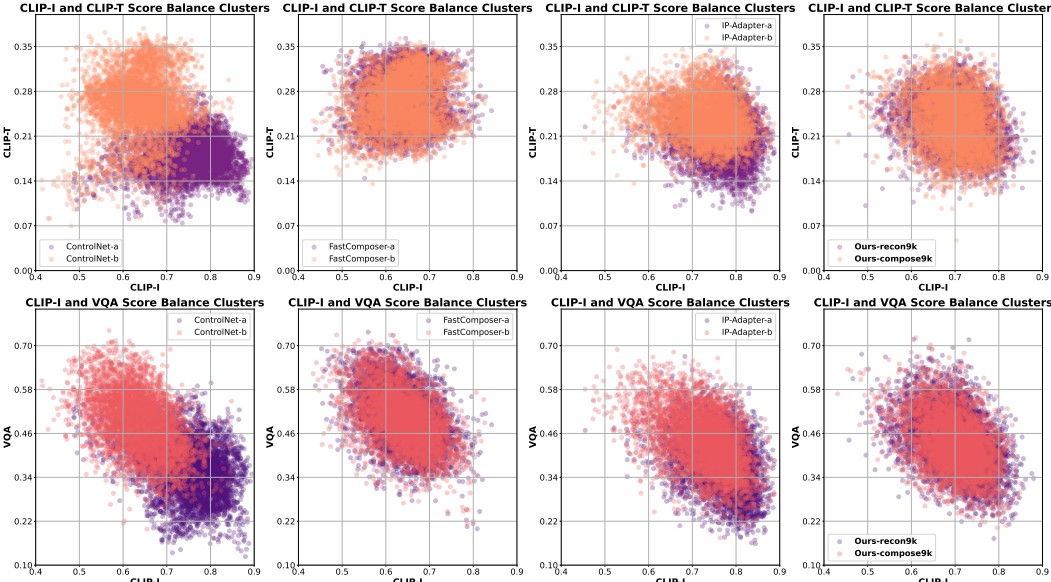

Figure 9: Detailed scatter plots for evaluating similarity-diversity balance. We observe that for benchmarking results, especially IP-Adapter, **-b** settings can improve the balance by shifting the clusters closer to the diagonal. Note that our proposed methods (rightmost column) achieves the best balance with the highest concentration, signifying the success of our proposed approaches. All models are pre-trained with the same length of GPU hours.

that it is moving more towards the y-axis and thus diagonal than **-a** settings (purples). Among the benchmarked methods, IP-Adapter achieves the best effect in improving the balance, while ControlNet models are shifted a bit excessively by **-b**, and FastComposer has smaller effect. We also include CLIP-I_CLIP-T and CLIP-I_VQA scatter plots for our RetriNet models (rightmost column) pre-trained on reconstruction and concept composition tasks for 9000 steps (same GPU hours). We cannot compare the improvement by a different task, but to provide a visualized comparison with previous methods such that our methods are more balanced (closer to the diagonal) and more concentrated with less imbalanced generation.

## A.4 MORE DETAILS OF NEW TASKS

We provide more details of our proposed reconstruction and concept composition tasks on RetriBooru, as well as training details using RetiNet. By default, we randomly choose $N = 4$ reference images from the "similar" entry of the target. During data processing, we can utilize head bounding boxes and masks to obtain face images and masks, as well as cloth images and masks, dividing into two concepts. **For reconstruction,** we simply pass whole-figure reference images with "figure" captions. **For concept composition,** we randomly choose $N$ reference images, and crop each image into "face" or "clothes" concepts with equal probability. While cropping a face image is straightforward, we obtain a cloth image by cropping out the region above the lower horizontal border of the face bounding box ($\leq$ `y_max`). In rare occasions where `y_max` $> \frac{2}{3} \cdot$ `height`, i.e. character in the reference image is upside-down or is a face close-up, we leave the reference image unchanged and caption with "figure" in concept composition task. **For multitask,** each batch has the equal probability to be processed in the same fashion as reconstruction or composition during data processing.

**RetriNet details.** We follow the official implementation of ControlNet (Zhang et al., 2023) and modify it into RetriNet. Specifically, we add additional cross attention layers right before zero-convolution layers, as illustrated in Figure 4. We implement our cross attention layers as shown in Code 1, with proper reshaping and layer normalization. Here, `control` is the output from our retrieval block and `hs` the output from U-Net encoder. We pass `control` as KV and `hs` as Q and output the final `hs` to the U-Net decoder.

```python
class ControlCrossAttention(nn.Module):                                          1
    def __init__(self, in_channels, num_heads):                                  2
        super().__init__()                                                       3
        self.norm0 = nn.LayerNorm(in_channels)                                   4
        self.norm1 = nn.LayerNorm(in_channels)                                   5
        self.norm2 = nn.LayerNorm(in_channels)                                   6
        self.gelu = nn.GELU()                                                    7
        self.cross_attention = nn.MultiheadAttention(in_channels,                8
                                    num_heads, batch_first = True)               9
        self.linear = nn.Linear(in_channels, in_channels)                       10
                                                                                11
    def forward(self, hs, control):                                             12
        control = control.reshape(hs.shape[0],                                  13
                control.shape[0]//hs.shape[0]*control.shape[1],                 14
                control.shape[2])                                               15
        hs = self.norm0(hs)                                                     16
        control = self.norm1(control)                                          17
        hs = self.cross_attention(hs, control, control)[0] + hs                18
        hs = self.gelu(self.linear(self.norm2(hs))) + hs                       19
        return hs                                                              20
```

Code 1: Python implementation of the cross attention layers that connect retrieval block to the U-Net.

**Training details.**    In addition to training settings specified in Section 5.2, we set up a cosine learning rate schedule with a initial rate$= 1^{-4}$, and restart the schedule every 9000 steps without decay. On target image, we use face image and mask for additional update and refine the generation, as specified in the training objective in Equation 3. Note that since our objective is still whole-figure generation, we need to choose a small coefficient $\lambda \approx 0.1$ for face loss, because equal weight would encourage the model to focus on learning faces (which are more stable than body parts), and generate more close-up images or more counterfactual body parts.

## A.5    ANIME RELATED WORK

Anime-based products have a lot of demand in the industry, where most social media platforms and image editing apps offer many popular anime related effects and filters. It is also a popular test-ground for many computer vision tasks, such as statistical image generation (Noguchi & Harada, 2019), subject-driven generation (Hua et al., 2023), facial recognition (Naftali et al., 2022), model watermarking (Qiao et al., 2023), multi-modal learning (Yi et al., 2023), transfer learning in GANs (Mangla et al., 2020), knowledge distillation (Cui et al., 2023), text detection (Del Gobbo & Matuk Herrera, 2020), image super-resolution (Dai et al., 2019), face detection (He et al., 2019)), etc.

## A.6    LIMITATIONS AND FUTURE MAINTENANCE

There are a few limitations of our work. First, we are limited by computation resources such that we could not provide further converged benchmarking training with optimal hyper-parameters. While our experiment results suffice to demonstrate the contribution of our proposed new training pipeline enabled by RetriBooru, we look forward to benchmark our dataset with more methods and exploit more ways of utilizing the rich annotations. Second, the performance of our annotations, including clothing identities, depends on the off-the-shelf models. While we are confident about the segmentation and detection quality since those models are pre-trained on a larger body of Danbooru dataset (Anonymous et al., 2022), clustering clothing identities depends on the generation quality of the chosen VQA model. With a more advanced open-source VQA model in the future, we would love to upgrade our annotations with more detailed question inputs. This also applies to SWD(VQA,CLIP-I), where answers to longer, more detailed questions contain more irrelevant responses. Furthermore, we will expand our dataset with the growing number of posts on Danbooru, and include more annotations of human-related concepts (e.g., hands and shoes).