# OpenReview forum: "RetriBooru: Leakage-Free Retrieval of Conditions from Reference Images for Subject-Driven Generation"
_ICLR.cc/2025/Conference — ICLR 2025 Conference Withdrawn Submission_

### Official Review · Reviewer_5uJE · 2024-10-29

**Soundness:** 3
**Presentation:** 2
**Contribution:** 2
**Rating:** 5
**Confidence:** 4

**Summary:**

The paper explains a method for creating a Danbooru-based dataset categorized by characters and concepts, and tests the impact of training with various reference images that align conceptually but differ visually from the target image. It also introduces metrics for managing diversity and similarity, presents a new task for combining concepts, and establishes a benchmark network for training.

**Strengths:**

- Originality:
The paper demonstrates some originality, which is reflected in the collection and validation of the effectiveness of avoiding target leakage on the anime dataset, as well as in the proposal of balancing metrics.
- Quality:
The experiments and analyses appear to be well-structured and comprehensive. The methodology seems sound, and the experiments have been designed to sufficiently validate the proposed approach.
- Significance:
The conclusions such as avoiding target leakage are reasonable and carry a certain level of significance.

**Weaknesses:**

- Limited Dataset Application: After validating the conclusions, it seems that the dataset can only be applied to customization or conditional encoder training within the anime domain.
- Insufficient Innovation in Concept Combination Task: The proposed concept combination task, which simply combines anime faces and clothing into a single image, lacks significant innovation or meaning. The visual results, such as in Figure 5 (prompt 1, second row), show relatively ordinary combinations, with low similarity between faces or clothing and the reference images.
- Vague Qualitative Results: The qualitative comparison results are somewhat ambiguous. It would be better to specify exactly what aspects are improved or to annotate the specific areas of improvement in the images.
- Need for More Experimental Results in Domain Generalization: Simple qualitative comparison images between two groups are insufficient. Therefore, conducting generalization tests for different conclusions, and even providing quantitative metrics, would be more convincing.
- Clarity in Explanation and Writing: Some parts of the explanation and writing could be clearer. For instance, the proposed balancing metric (Equation 2), which appears to be missing a left parenthesis, lacks clear and understandable explanations. Illustrating how this metric addresses ‘a worse trade-off’ mentioned in your paper would clarify its effectiveness.
- Domain Difference In Section 3.1:  Your evaluation set includes 40 in-domain characters and 10 out-of-domain characters, but this distinction is not further explained in the results. I am curious whether the domain difference has any impact on the metrics. Are there any qualitative results you can show?

**Questions:**

- Can you annotate the improvements in your qualitative results compared to the baseline in the images? Or can you describe in words what the advantages are?
- It would be better to show more results of different conclusions under domain generalization. For example, what is the impact of different text drop rates?
- In the current visualizations, are the combined concepts all from the same character's face and clothing, just with different reference images? Are there any unconventional combinations, such as one character's face with another character's clothing?
- Can you demonstrate the differences in metrics between the 40 in-domain and 10 out-of-domain characters mentioned in Section 3.1?

---

### Official Review · Reviewer_8Te1 · 2024-11-02

**Soundness:** 2
**Presentation:** 3
**Contribution:** 3
**Rating:** 5
**Confidence:** 4

**Summary:**

This paper propose RetriBooru Dataset, allowing training with reference images that are different from the target image but share the same identity. A new concept composition task is introduced and RetriNet is designed for this task. Finally, this paper introduces a new metrics named SWD to measure the similarity and diversity.

**Strengths:**

1. RetriBooru, this dataset is useful for future research.
2. The new concept composition task proposed in this paper is interesting and worth study.

**Weaknesses:**

1. In the comparison of qualitative results Figure3, apart from prompt2 with scale=0.75 where the advantage of method B is evident, I cannot see clear advantages of IP-Adapter-b over IP-Adapter-a in other case.
2. In Figure7, I have the same opinion for the result generalizing to human face. ie, I don't think training pipeline of this paper keeps a better  identity-diversity balance, and the human ID is worse than other methods. Although the author claims the result is limited by V100, it is hard to distinguish whether the training pipeline or GPU limit the performance. And the effectiveness of this method on real datasets is not convincing.

**Questions:**

1. In Figure4, why some area in noise target image is masked?
2. In Figure4, two faces and two clothes are input as reference concepts. When inference, is it necessary two faces and two clothes?
3. Is the dataset will be public?

---

### Official Review · Reviewer_yR3Y · 2024-11-02

**Soundness:** 2
**Presentation:** 2
**Contribution:** 2
**Rating:** 3
**Confidence:** 4

**Summary:**

The paper proposes a dataset termed RetriBooru, which aims to solve the leakage of spatial information.
The dataset is multi-level and same-identify with some reference images with different attributions.
The authors use the dataset to train baseline models for performance verification.

**Strengths:**

The paper contributes a new dataset to solve the information leakage problem by using part images with different attributions or descriptions.

**Weaknesses:**

1. I question whether the motivation is reasonable. The authors suggest that the similarity between the reference image and the target image can easily lead to spatial information leakage. Although they propose using partial images with different attributes to address this issue, these partial images can also contain spatial information. Furthermore, collecting such partial images for general domains is challenging, and ensuring their accurate and comprehensive annotation may not be fully achievable.

2. The comparisons presented in Table 1 are not clearly delineated. Based solely on the information in the table, readers may struggle to identify any significant differences between this dataset and others.

3. The authors have exclusively utilized anime images to construct the dataset, which may limit its generalizability. Anime images often feature simplistic backgrounds and restricted attributes. I wonder if this could contribute to the poor performance observed in Figure 7 (prompt 2).

4. In Figure 4, I have concerns regarding the comprehensiveness of the annotation for the partial images. Would providing attributes from the entire area enhance generation performance? Additionally, how can we effectively control the details of the entire face? Should the dataset be updated to incorporate these insights? Lastly, how should the granularity of the annotations be defined?

**Questions:**

1. Does the dataset specify the annotation level? From the information provided in the paper, I am unable to find any evidence regarding this.

2. The motivation concerning information leakage is not substantiated throughout the paper. Could you provide relevant evidence to demonstrate the extent to which information leakage has been reduced?

3. A partial image may exhibit different angles. How is the selection of partial images with varying attributes determined?

4. How can we mitigate the influence of partial images that are not addressed in the textual description?

---

### Official Review · Reviewer_Fi41 · 2024-11-03

**Soundness:** 2
**Presentation:** 3
**Contribution:** 2
**Rating:** 5
**Confidence:** 3

**Summary:**

This paper proposes a large-scale dataset RetriBooru for training subject-preserving generation model, for each identity, it has multiple image include various scale and cloth. They introduce a new composition task, where the conditioning encoder learns to retrieve different concepts from several reference images. For evaluation, they propose new metrics named Similarity Weighted Diversity (SWD), aim to measure the alignment between similarity and diversity.
Their main contribution are in following aspects:
1.	Proposing an anime dataset RetriBooru for anime character preserving generation with desired cloth with diverse poses.
2.	Proposing a new training pipeline for subject-driven generation by using a reference image different from the target but with the same identity.

3.	Proposing a new concept composition task, which learns to retrieve specific aspect information or various concepts from different reference images, and provide baseline results with our baseline architecture RetriNet.

4.	Proposing new metrics (SWD) to evaluate referenced image generation methods.

**Strengths:**

Authors propose Similarity Weighted Diversity (SWD) metrics to jointly measure the image similarity and text alignment which can be a new type metric to evaluate identity-preserving of existing subject-preserving generation models.
Author propose new concept composition task which takes different reference images from same identity to generate new image in same identity. It enables a flexible pose, action generation but preserving the identity and cloth, etc. from reference images.

**Weaknesses:**

The paper title and several parts in the main text of paper mention “leakage-free” and “reduce leakage”. However, there is no supported related experiments conducted by authors.
Authors propose -b training scheme that uses a same-identity but different image from target as the reference image. However, the definition target is not clear and it is not unique for each identity. Training scheme -a could also be one case of -b training setting.

-b training scheme doesn’t ensure a general good performance in CLIP-I and CLIP-T scores. For example, -b training scheme decrease CLIP-I on ControlNet and IP-Adapter and decrease CLIP-T on Kandinsky-0.1, Kandinsky-0.2.

**Questions:**

What is the solid example that can showcase the propose method can reduce leakage of target image? We suggest authors provide quantitative or qualitative analysis of reduced target image leakage. For example, providing side-by-side comparisons of generated images with and without your proposed method and highlight the areas that contain unwanted information.

In the Evaluation part, why set 384 resolution for each model? The size 384 is relative resolution compared the default resolution in reference stage of some models. For example, IP-Adapter is trained at a resolution of 512x512 and it has degraded subject-preserving ability when generating low resolution image. We suggest justify the choice of 384 and discuss how this choice may affect comparisons to models typically used at higher resolutions by conducting ablation study.

---

### Note · Authors · 2024-11-20

**Comment:**

We sincerely thank the reviewers for their interests, the acknowledgement on some aspects of our work, as well as their detailed, valuable feedback. However, we feel that addressing all of the concerns would lead to a large revision that is worth resubmission, with more emphasis on the target leakage and the applications and insights on newly proposed task. Therefore, we decide to withdraw this submission at this point. Thanks for the time and advice.

Sincerely,
7725 Authors

**Withdrawal Confirmation:**

I have read and agree with the venue's withdrawal policy on behalf of myself and my co-authors.